# Parvovirus B19 in Pregnancy—Do We Screen for Fifth Disease or Not?

**DOI:** 10.3390/life14121667

**Published:** 2024-12-16

**Authors:** Andrei Mihai Malutan, Cristina Mihaela Ormindean, Doru Diculescu, Razvan Ciortea, Renata Nicula, Daria Pop, Carmen Bucuri, Roman Maria, Ionel Nati, Dan Mihu

**Affiliations:** 12nd Department of Obstetrics and Gynecology, ‘Iuliu Hatieganu’ University of Medicine and Pharmacy, 400006 Cluj-Napoca, Romania; amalutan@umfcluj.ro (A.M.M.); ddiculescu@yahoo.com (D.D.); r_ciortea@umfcluj.com (R.C.); renatanicula@elearn.umfcluj.ro (R.N.); dariagroza@elearn.umfcluj.ro (D.P.); cbucurie@elearn.umfcluj.ro (C.B.); nati.ionel.daniel@elearn.umfcluj.ro (I.N.); dan.mihu@elearn.umfcluj.ro (D.M.); 2Clinical Department of Surgery, Constantin Papilian, Emergency Clinical Military Hospital, 22 G-ral Traian Mosoiu, 400132 Cluj-Napoca, Romania

**Keywords:** pregnancy, parvovirus, infection, fifth disease

## Abstract

Parvovirus B19 (B19V) infection is the cause of erythema infectiosum, or the “fifth disease”, a widespread infection, potentially affecting 1–5% of pregnant women, in most cases without significant damage to the pregnancy or fetus. It follows a seasonal variation, with a higher prevalence in temperate climates, mainly in late winter and early spring. Women at increased risk include mothers of preschool and school-age children, and those working in nurseries, kindergartens, and schools. Vertical transmission occurs in 33% to 51% of cases of maternal infection. Parvovirus infection is an important cause of fetal perinatal infection resulting in increased morbidity through the development of fetal anemia, heart failure, and non-immune hydrops. A comprehensive literature review was conducted using PubMed, Cochrane Library, and Google Scholar, focusing on publications from the last 10 years and prioritizing studies related to parvovirus B19 infection in pregnancy. We summarized the existing data in the literature on the effects of parvovirus B19 infection during pregnancy and weighed if there is a need for screening in pregnant patients. Routine screening for parvovirus B19 infection can be considered in communities where infection is common, there is occupational exposure, or during endemic periods, with the reason being that accurate identification and treatment of fetuses affected by congenital B19V infection have been shown to improve perinatal outcomes.

## 1. Introduction

Infections during pregnancy have become a serious public health problem worldwide. Some maternal infections, contracted immediately before or during pregnancy, can be transmitted to the fetus during gestation (congenital infection), labor and delivery (perinatal infection), or through breastfeeding (postnatal infection). The agents responsible for these infections can be viruses, bacteria, protozoa, or fungi [1,2]. Although much less common than bacterial infections, the occurrence of congenital and perinatal viral infections can have serious consequences for the newborn, some with important sequelae during childhood and later in adult life. Among the viruses most associated with the development of congenital infections are cytomegalovirus (CMV), Herpes simplex 1-2 (HSV 1-2), Herpes virus 6 (HHV-6), Varicella zoster virus (VZV), and Rubella virus (RuV). In addition, other important infections associated with pregnancy include hepatitis B and C viruses (HBV and HCV), human immunodeficiency virus (HIV), parvovirus B19 (B19V), or non-polio enterovirus (EV). Recently, new micro-organisms have emerged, with the coronavirus responsible for severe acute respiratory syndrome (SARS-CoV-2) and Zika virus (ZIKV), whose effects on pregnancy are not yet fully known [1,2,3].

For the most common viral infections, screening tests have been introduced, such as the TORCH complex, an acronym that includes *Toxoplasma gondii*, RuV, CMV, and herpes viruses, all potential causes of congenital infections. It was first introduced in 1970, and subsequently, other acronyms have been introduced to cover a wider range of infections [1,2,3,4]. In addition to these entities included in the screening tests, there are other viruses whose notable effects impact both the course of pregnancy and the fetus, and of these, an important agent is parvovirus B19 [1,2].

Parvovirus B19 infection is the cause of erythema infectiosum, the “fifth disease” or “slapped cheek disease”, a common childhood illness. The term “fifth disease” was employed because parvovirus B19 infection has similarities with the existing classic diseases, the four childhood rashes: measles, scarlet fever, rubella, and “fourth disease” or Dukes’ disease [5,6].

The prevalence of parvovirus B19 infection varies depending on the population and region studied. Globally, the majority of individuals have been exposed to parvovirus B19 by older adulthood, as evidenced by a seroprevalence rate of approximately 85%. In Europe, the proportion of women of childbearing age who are susceptible to parvovirus B19 infection varies. Studies have estimated susceptibility rates of 26% in Belgium, 38% in England and Wales, 43.5% in Finland, 39.9% in Italy, and 36.8% in Poland. While there is currently no prevalence study for Romania, rates are expected to be similar to those of other European countries, with seroprevalence increasing with age. Studies in Romania’s neighboring countries, including Hungary, Bulgaria, and Serbia, have reported seroprevalence rates ranging from 30% to 60% in adults [1,2,3,4,5,6,7,8].

The clinical features of human parvovirus B19 infection were described in the 1980s by Pattison in children with sickle cell disease who developed transient aplastic anemia. Later, Anderson et al. found that this virus causes erythema infectiosum, particularly in children [6,7]. Parvovirus B19 infection is widespread, potentially affecting 1–5% of pregnant women. While most cases do not result in significant damage to the pregnancy or fetus, this infection can lead to serious complications such as fetal anemia and reticulocytopenia, particularly during the second trimester. These hematologic disturbances arise from the virus’s disruption of red blood cell production and can have significant consequences for the developing fetus. The prevalence of infection is higher during epidemics, with infection rates ranging from 3% to 20%, with maternal seroconversion rates between 3 and 34%. This infection follows a seasonal variation, with a higher prevalence in temperate climates, mainly in late winter and early spring. Outbreaks usually occur annually, with larger outbreaks every four to five years, which can last up to six months, during which time, children and their contacts are more at risk of infection. Most cases of infection in pregnant women seem to occur in late spring and summer. Women at increased risk include mothers of preschool and school-age children, and those working in nurseries, kindergartens, and schools [5,6,7,8].

Assessing immunity to B19V during early pregnancy may be beneficial for this population, as approximately 50% to 75% of women of reproductive age have detectable antibodies, indicating prior infection and lasting immunity. Among pregnant women with acute B19V infection, around 30% report polyarthralgia, and 30–40% experience a rash. Symptoms, when present, are typically brief, lasting only 2–5 days. Overall, 30–50% of acutely infected pregnant women remain asymptomatic. Vertical transmission occurs in approximately 33% to 51% of maternal cases. While most cases of acute B19V infection during pregnancy do not adversely affect outcomes, this infection is a significant cause of fetal infection and increased morbidity, as it may lead to fetal anemia, heart failure, and non-immune hydrops. The risk of anemia and hydrops complications ranges from 3.9% to 11.9%, with peak incidence occurring between 20 and 24 weeks of gestation. These complications are critical contributors to increased fetal and neonatal mortality, particularly during the second trimester [6,7,8,9].

*Parvovirus B19* is a small (20–25 nm), icosahedral, non-enveloped virus containing linear single-stranded DNA. The capsid contains the two structural proteins (VP1 and VP2) and is involved in many functions of the virus life cycle. The capsid surface mediates specific binding to host cell receptors, followed by internalization mediated by its phospholipase A2 (PLA2) activity, penetration into the nucleus, and recognition and avoidance of the host immune response [1,5,9,10].

While global prevalence data exist, there is a lack of comprehensive studies investigating the specific prevalence and epidemiology of parvovirus B19 infection in Romania. This lack of data poses challenges in developing targeted prevention and screening strategies tailored to the Romanian population. Therefore, conducting large-scale national studies is crucial to determine the seroprevalence of parvovirus B19 antibodies in pregnant women and women of childbearing age in Romania, identify high-risk populations and geographical areas with higher prevalence, evaluate the impact of parvovirus B19 infection on maternal and fetal health outcomes in the Romanian context, and assess the cost-effectiveness of different screening strategies in the Romanian healthcare system.

Gathering this valuable epidemiological data will enable evidence-based decision-making regarding the implementation of targeted screening and prevention programs in Romania, ultimately improving maternal and child health outcomes.

Given the potential risks associated with parvovirus B19 infection during pregnancy, this study aimed to provide a comprehensive overview of the current literature on the effects of this infection. Furthermore, we sought to evaluate the necessity of screening for parvovirus B19 in pregnant patients, considering the potential benefits and drawbacks.

## 2. Materials and Methods

This study employed a narrative review methodology. For data collection, we searched the available online databases, PubMed, Cochrane Library, and Google Scholar, encompassing publications from January 2014 to December 2023, using the following keywords: Parvovirus B19 infection AND fetal transmission, Parvovirus B19 infection AND (congenital infection OR pregnancy OR fetal effects), Parvovirus B19 infection AND screening, Parvovirus B19 infection AND diagnosis, Parvovirus B19 infection AND management, Parvovirus B19 infection AND (non-immune hydrops OR fetal anemia OR miscarriage OR fetal death) (Figure 1). We also reviewed the references of the identified studies to enrich our knowledge on this topic. All data included in this narrative review have been prepared following the criteria in the PRISMA checklist. We have included cohort studies and case-report articles on pregnant women with confirmed parvovirus B19 infection. For the sections on ‘Taxonomy’ and ‘Epidemiology and transmission’, we included seminal papers and recent reviews to provide a comprehensive overview of the topic. Duplicate records identified across databases were removed. We included relevant non-peer-reviewed sources, such as government reports and conference proceedings, only when they provided unique information not available in peer-reviewed literature.

## 3. Taxonomy

Parvovirus B19 belongs to the family Parvoviridae, which is subdivided into Parvovirinae and Densovirinae based on their genomic organization, virion structure, and host range, as described by the International Committee on Taxonomy of Viruses (ICTV). The Parvovirinae group includes the Erythroparvovirus genus, which requires the presence of erythroid cells to replicate. Parvovirus B19 belongs to the genus Erythrovirus and affects only the human species. Currently, three genotypes are known, with genotype 1 being the most widespread. While all three genotypes can cause similar clinical manifestations, they may exhibit variations in their geographical distribution and prevalence. Genotype 1 is the most common cause of infection worldwide, while genotype 2 is more frequently observed in older adults. Genotype 3 is less common and is found mainly in Africa and South America. Some studies suggest potential differences in their virulence and association with specific clinical outcomes, but further research is needed to fully elucidate these distinctions. The three genotypes exhibit a 13 to 14 percent difference in their nucleotide sequences. As expected, the divergence at the amino acid level among the three genotypes is significantly smaller than that observed at the nucleotide level [1,5,6].

## 4. Epidemiology and Transmission

Parvovirus B19 infection is globally prevalent, with seropositivity rates demonstrating a distinct pattern of change throughout life. In early childhood, seroprevalence is relatively low, with approximately 15% of preschool children exhibiting evidence of past infection. This rate increases significantly as children reach school age and interact more with others, reaching around 50% in young adults. By older adulthood, the majority of individuals have been exposed to parvovirus B19, as evidenced by a seroprevalence rate of approximately 85%. This age-related pattern of seropositivity has important implications for risk assessment. Women of childbearing age who lack immunity to parvovirus B19 are at increased risk of acquiring the infection during pregnancy, particularly if they have young children or work in settings with frequent exposure to children. Understanding the seroprevalence dynamics across different age groups can aid in identifying high-risk populations and implementing appropriate preventive measures, such as vaccination or minimizing exposure during outbreaks. Between 25% and 50% of those infected remain asymptomatic [5,6,7,8]. Incidence peaks typically occur in late winter and early spring, with periodic outbreaks every few years. B19V primarily spreads through respiratory droplets but can also be transmitted vertically from mother to fetus via the placenta or through blood products during transfusion. The virus exhibits a tropism for erythroid progenitor cells in the bone marrow, with globoside playing a crucial role in facilitating viral entry and endosomal escape. Vertical transmission of the infection from mother to fetus occurs when the virus crosses the placenta during pregnancy. This transmission is most frequent in the first and second trimesters, with a peak around 13 to 20 weeks of gestation. The virus has a particular affinity for erythroid progenitor cells, which are the precursors to red blood cells. This is because these cells possess a specific receptor, the globoside or P blood group antigen, to which the parvovirus B19 binds. This binding allows the virus to enter and infect the erythroid progenitor cells, disrupting the production of red blood cells and potentially leading to fetal anemia [7,9,11].

## 5. Clinical Presentation

Clinical presentations of parvovirus B19 infection vary significantly, with some individuals remaining asymptomatic and others experiencing prodromal symptoms. Among non-pregnant women, approximately 50% acquire parvovirus B19, and 30–50% of infected pregnant women remain asymptomatic. In certain cases, infection is followed by a symptomatic period, and in immunocompromised patients, it can become chronic, leading to long-term complications, likely due to the persistent viral presence in erythroid progenitor cells and potentially other susceptible cell types. The incubation period is typically around 14 days post-exposure but may extend to 3 weeks. The classic presentation of erythema infectiosum includes a maculopapular rash with a reticulated, lacy pattern, primarily on the face, often resembling slapped cheeks, though it may also involve the trunk or extremities. The immune response and symptom progression are biphasic, with a second phase occurring approximately 17–18 days post-infection, characterized by rash, pruritus, and joint pain. Another notable feature is peripheral polyarthropathy, mainly affecting the knees or hand joints; in pregnant patients, this may be the sole manifestation, occurring in up to 50% of cases, and may persist for several weeks to months. The onset of arthritis aligns with IgM antibody detection, typically 10–12 days post-primary infection, which remains positive for up to 3 months, while IgG antibodies persist for life [4,5,6,7,8,9,10].

In immunocompromised patients, parvovirus B19 infection can cause chronic anemia through bone marrow suppression. The resulting anemia is often accompanied by a low reticulocyte count (reticulocytopenia), as the virus disrupts red blood cell production. This disruption occurs because the virus has a strong affinity for erythroid progenitor cells in the bone marrow, which are responsible for generating red blood cells. Infection and subsequent replication within these cells lead to the formation of giant proerythroblasts (Figure 2), large, immature red blood cells characteristic of B19V infection. These abnormal cells are often destroyed before they can mature into functional red blood cells, further contributing to the anemia and reticulocytopenia. In some cases, the immune response to the virus may also contribute to red cell aplasia by attacking and destroying infected erythroid progenitor cells [12]. The infection has also been associated with the development of myocardial diseases such as myocarditis, dilated cardiomyopathy and left ventricular dilatation, ventricular cardiac dysfunction, or renal damage such as glomerulonephritis, although a causal link has not been definitively established [10,13].

## 6. Parvovirus B19 Infection During Pregnancy—Pathogenesis of Maternal–Fetal Impairment

Parvovirus B19 vertical transmission rate ranges from 17% to 30%, most seen between 9- and 20-weeks’ gestation [5,9]. The presence of the P antigen in trophoblastic tissue, along with other potential co-receptors, facilitates the transmission of infection to the embryo or fetus from the infected mother. Recent evidence suggests that globoside, in addition to its role in erythroid progenitor cells, may also be involved in viral entry within placental tissue [10]. The expression of this antigen in the placental tissue is dependent on gestational age, with increased immunoreactivity identified in the first trimester of pregnancy, as opposed to the third trimester where immunoreactivity is almost absent [9,10].

Once in the fetal circulation, tissues such as hematopoietic cells in the liver, myocardium, endothelial cells, platelets, megakaryocytes, and fibroblasts have been shown to express the viral P antigen receptor [10]. This receptor distribution aligns with the range of clinical manifestations observed in fetal infections [9]. The fetus is particularly susceptible due to the shorter lifespan of fetal red blood cells (50–75 days) [8]. Viral-induced cytotoxicity and apoptosis lead to a halt in fetal erythropoiesis, resulting in anemia, tissue hypoxia, myocarditis, cardiomegaly, pericardial effusion, and ultimately to hydropic or non-hydropic intrauterine fetal death (IUFD) [14].

Most fetuses infected with parvovirus B19 experience spontaneous remission of infection without adverse outcomes [10,14]. Effects of viral infection during pregnancy include the following (Table 1):Spontaneous abortion: parvovirus-infected fetuses’ spontaneous loss rate before 20 weeks of gestation is 13%, and after this gestational age, the rate drops to 0.5%. The reason for this difference is not known, but one large study suggests that it may be related to multisystem damage, which may appear even in the absence of anemia or hydrops [10,15].Non-immune hydrops, often recognized as the most prevalent and visible sign of congenital infection, primarily present as fetal hydrops. The likelihood of hydrops development is closely associated with the gestational age at which maternal infection occurs. If the infection arises within the first 12 weeks of gestation, the risk of hydrops ranges from less than 5% to approximately 10%. During the 13th to 20th week, this risk diminishes to 5% or lower, and after the 20th week, it further declines to 1% or less [9,10,16]. When hydrops occurs in the fetus, sonographic indicators typically include ascites, skin edema, pleural and pericardial effusions, and placental edema. Mechanisms implicated in hydrops formation involve fetal anemia, caused by a transplacental viral infection that affects red cell progenitors in the fetal bone marrow. The shorter lifespan of fetal red blood cells, particularly during the liver stage of hematopoiesis, exacerbates anemia, leading to hypoxia and increased cardiac output, potentially resulting in heart failure. Additional factors include fetal viral myocarditis, which can lead to cardiac failure, and compromised liver function due to both direct hepatocyte damage and indirect injury from hemosiderin deposition [5,7,9,10,17,18].Congenital abnormalities: currently, there is no evidence that parvovirus B19 infection increases the risk of congenital abnormalities in humans [5,9], although cases of central nervous system, craniofacial, musculoskeletal, and ocular abnormalities have been reported [5,7,8,10].Long-term neonatal effects: studies examining the long-term impact of maternal parvovirus B19 infection on infants indicate that most newborns do not experience lasting adverse effects, though further research is warranted [5,9]. In a study by Miller et al., no increased risk of adverse outcomes was observed in children of mothers with parvovirus infection during pregnancy, assessed at 1 year (182 children) and 7–10 years (129 children) [5]. Conversely, research by Nagel et al. found abnormal neurodevelopment in 5 of 16 infants who had undergone intrauterine blood transfusions for parvovirus B19 infection [5]. Parvovirus infection alone, in the absence of hydrops or significant fetal anemia, does not seem to be linked with long-term neurological issues; however, severe anemia and fetal hydrops may independently increase the risk of neurological sequelae. Brain imaging studies could be considered for neonates diagnosed with severe anemia and hydrops [5,8]. Additionally, parvovirus-related myocarditis can lead to severe dilated cardiomyopathy, sometimes requiring heart transplant surgery [5,9,10].

## 7. Parvovirus B19 Infection During Pregnancy—Diagnosis of Maternal–Fetal Infection

### 7.1. Diagnosis of Maternal Infection

Currently, routine antenatal serologic screening for parvovirus B19 immunity is not formally recommended. Testing is typically conducted when exposure is confirmed or when ultrasound findings indicate hydrops fetalis or unexplained fetal anemia. Maternal infection is diagnosed primarily through antibody detection, though PCR (Polymerase Chain Reaction) for viremia is also available. The presence of anemia and reticulocytopenia, especially in pregnant women with known exposure to parvovirus B19, should prompt consideration for antibody testing to confirm the diagnosis. These hematologic findings can serve as an early indicator of potential infection, allowing for timely intervention and monitoring to minimize adverse fetal outcomes [10,19,20]. Figure 3 outlines the role of serology in guiding diagnostic and clinical management for pregnant patients. Pregnant women exposed to parvovirus B19 should undergo serologic testing promptly after exposure to assess the risk of acute infection [9,10,21]. Figure 4 shows the kinetics of IgM and IgG antibodies against B19v infection, and IgM antibodies are the first to appear after pB19v infection. They are detectable within 1–2 weeks after exposure and usually peak around 2–3 weeks. IgM levels then decline and become undetectable within a few months, though they can sometimes persist longer, IgG antibodies appear later than IgM, usually around 2–3 weeks after infection. IgG levels rise gradually and then plateau, providing long-term immunity against pB19v [10,19,20,21]. Also, while establishing the diagnostics of maternal infection, IgG avidity can be used, which refers to the strength of the bond between IgG antibodies and the pB19V antigen (Figure 4). It increases over time as the immune response matures. Measuring IgG avidity can help differentiate between recent and past infection. High avidity suggests past infection, while low avidity suggests recent or acute infection [10,17,19,20,21]. If IgG antibodies are present and IgM is negative, the patient is immune and at no risk of developing infection [10,20]. If both IgG and IgM are negative, the patient is susceptible and should be retested for seroconversion 2–4 weeks later, as asymptomatic infections are possible [5,10]. A positive IgM result indicates a recent infection, but a negative result should be interpreted based on the time since exposure. Maternal serology may be unreliable if performed within 7 days of exposure, as IgG and IgM may both be negative at this early stage [8,10]. IgM antibody sensitivity for parvovirus B19 infection is estimated to be 63–70% between 8 and 12 weeks after maternal infection. This means that in a group of pregnant women who were infected with parvovirus B19, only 63–70% will test positive for IgM antibodies within this timeframe. The remaining women will have a false negative result, meaning they are infected despite the negative test [9,10,19]. False negative IgM results can occur due to several factors including timing of the test (IgM antibodies may not be detectable in the very early stages of infection—within 7 days of exposure), waning antibody levels (by the time fetal hydrops develops, IgM antibody levels may have already decreased to undetectable levels), and individual variation (some individuals may have a weaker or delayed IgM antibody response). Due to the possibility of false negatives, a negative IgM result cannot definitively exclude fetal infection if maternal exposure occurred 2–3 months earlier. Therefore, clinicians should consider the time since exposure (if exposure occurred recently, PCR testing may be more sensitive in detecting early infection), clinical presentation (symptoms such as rash, joint pain, or ultrasound findings suggestive of fetal infection warrant further investigation even if IgM is negative), and repeat testing (if IgM is negative and suspicion remains high, repeat serologic testing 2–4 weeks later may be necessary to assess for seroconversion). Understanding the limitations of IgM antibody sensitivity is crucial for accurate diagnosis and appropriate management of parvovirus B19 infection during pregnancy [6,10,21].

Detection of B19 DNA in maternal plasma through PCR can enhance diagnostic accuracy in early infection stages. B19 viremia can be identified 5 to 10 days post-exposure, before any detectable serological changes, and in immunocompetent individuals, viremia may be present up to 7 days before symptom onset. However, PCR results are generally negative after maternal symptoms appear [10,22]. In a study of 72 pregnant patients with pregnancies complicated by parvovirus B19 infection, IgM serology correctly diagnosed 94.1% of cases, while PCR accurately diagnosed 96.3% [9,10,23]. For pregnant patients with recent parvovirus exposure and initial negative serology, PCR screening may maximize diagnostic sensitivity. Nonetheless, PCR is not routinely used due to its limited clinical utility; maternal diagnosis is usually not urgent, and confirmation through serologic testing is preferred [8,10,17,21,22].

In some cases, a bone marrow aspirate may be performed to examine the bone marrow cells directly. The presence of giant proerythroblasts, which are large, immature red blood cells with characteristic features, is considered pathognomonic (diagnostic) for B19V infection. A peripheral blood smear can also provide clues to B19V infection. Although not as specific as bone marrow aspirate, it may reveal abnormalities in red blood cell morphology, such as anisocytosis and poikilocytosis, which can be suggestive of the infection [22,24,25].

No antiviral or immunoglobulin therapy exists for B19V infection, nor is there a treatment to prevent fetal infection following maternal exposure. Consequently, managing congenital B19V infection focuses on preventing severe fetal complications related to anemia [9,10].

### 7.2. Diagnosis of Fetal Infection

Detection of B19V DNA in amniotic fluid obtained through amniocentesis is the most common method used to determine the cause of unexplained fetal hydrops [5,9,10]. In this scenario, maternal serology is informative only if both IgG and IgM are negative, effectively ruling out maternal and fetal B19V infection. Although PCR testing for parvovirus B19 can be conducted on both fetal cord blood and amniotic fluid, amniocentesis is generally preferred for the following reasons:Higher detection rate: Amniotic fluid provides a higher detection rate for B19V DNA compared to fetal cord blood. This means that the virus is more likely to be identified in the amniotic fluid, leading to a more accurate diagnosis.Hess invasive: Amniocentesis is considered less invasive than cordocentesis (fetal blood sampling). It involves inserting a thin needle through the mother’s abdomen and uterus to collect a small sample of amniotic fluid, while cordocentesis requires accessing the umbilical cord, which carries a slightly higher risk of complications.Lower risk of fetal loss: The risk of fetal loss associated with amniocentesis is generally lower than that of cordocentesis. While amniocentesis carries a small risk (estimated to be around 1 in 300 to 1 in 500), cordocentesis has a slightly higher risk, ranging from 1% to 2% [5,7,9,10].

Therefore, the higher detection rate, lower invasiveness, and reduced risk of fetal loss make amniocentesis a preferred method for detecting B19V DNA and diagnosing fetal infection [10,19,24]. Fetal serology is seldom employed diagnostically because the fetal immune system is immature, resulting in an unreliable IgG/IgM response [10,24]. Detecting specific IgM in fetal blood has a sensitivity of only 29%, and percutaneous fetal blood sampling carries a 1% risk of fetal loss. Patients who seroconvert with normal fetal ultrasound findings are typically managed non-invasively, with ultrasound monitoring for signs of fetal anemia, rather than undergoing amniocentesis [24].

If maternal infection with parvovirus B19 is diagnosed before 20 weeks of gestation, expert evaluation is required, as infection in the first half of pregnancy confers the highest risk for developing fetal anemia. Documentation of fetal anemia is performed using ultrasound examination at 1–2-week intervals for at least 12 weeks from the time of diagnosis of maternal infection [10,16,21,24]. Anemia is detected using Doppler measurement of the peak systolic velocity (PSV) in the mid-cerebral artery (MCA), which is accurate in identifying anemia using a PSV threshold of >1.5 MoM from 18 weeks of amenorrhea [10,16,20,24]. The risk of anemia is higher in fetuses with a pre-transfusion PSV of ≥1.5 MoM, although lower thresholds (e.g., 1.29 MoM) have greater sensitivity for lower specificity [9,10,21]. A multicenter study of MCA PSV that included 32 at-risk fetuses (due to maternal parvovirus infection) demonstrated that parvovirus infection can be detected noninvasively using Doppler ultrasonography. Cordocentesis was performed in 16 of 17 fetuses with MCA PSV values >1.5 MoM and confirmed the diagnosis of anemia [10]. The classic reference charts for MCA PSV developed by Mari et al. give values from 18 weeks [9,10], but there are adaptations for lower gestational ages. The frequency of assessing at-risk pregnancies is higher at weekly examinations once the option of fetal transfusion becomes a necessary therapeutic method. If hydrops or signs of fetal anemia occur following PSV MCA monitoring, the patient should be referred to a specialist capable of performing fetal blood sampling and intrauterine transfusion [10,21].

Early indicators of non-immune hydrops fetalis include the presence of ascites and cardiomegaly. In more advanced stages, generalized edema, pericardial effusion, and a thickened, edematous placenta may be observed [8,10,25]. These placental changes are believed to contribute to the occasional occurrence of maternal “mirror syndrome,” a condition resembling pre-eclampsia characterized by edema, hypertension, proteinuria, and maternal anemia, reflecting the fetal changes [9,10]. Additional sonographic signs associated with parvovirus B19 infection include echogenic bowel, meconium peritonitis, increased nuchal translucency in the first trimester, abnormalities in amniotic fluid, and myocardial dysfunction [8,10].

### 7.3. Implications of Antibody Determination in the Context of Anemia with Reticulocytopenia

In assessing the implications of antibody determination specifically when anemia with reticulocytopenia is present, several factors warrant careful consideration. These include the cost-effectiveness of testing, the potential benefits of alternative screening strategies, and the potential consequences of delaying antibody testing until these hematologic manifestations are evident [1,4,10,15].

Antibody testing, while essential for confirming parvovirus B19 infection, can be expensive. In resource-limited settings like Romania, where healthcare budgets may be constrained, indiscriminate antibody testing for all pregnant women may not be feasible. Therefore, a more targeted approach is necessary to ensure that testing resources are used judiciously [1,4,9,21].

An initial screening strategy using complete blood count (CBC) and reticulocyte count could be a cost-effective alternative. This approach can help identify pregnant women with anemia and reticulocytopenia, who are then considered high-risk for parvovirus B19 infection. These individuals can then undergo more specific antibody testing to confirm the diagnosis. This tiered approach can help optimize resource allocation while ensuring that high-risk individuals are identified and managed promptly [1,8,10].

Delaying antibody testing until anemia with reticulocytopenia develops can have potential consequences. Early diagnosis of parvovirus B19 infection is crucial for initiating timely interventions and monitoring to minimize adverse fetal outcomes. Anemia and reticulocytopenia can indicate advanced infection, potentially leading to delayed treatment and increased risk of complications. Early detection through antibody testing allows for prompt management, which can improve maternal and fetal outcomes [1,4,8,9,15].

Given the cost implications and the potential consequences of delayed diagnosis, a balanced approach is recommended. In settings like Romania, an initial screening strategy using CBC and reticulocyte count can be considered. This can be followed by targeted antibody testing for those identified as high-risk based on the initial screening. This strategy can help optimize resource utilization while ensuring timely diagnosis and management of parvovirus B19 infection in pregnant women.

## 8. Pregnancy Management

Intrauterine transfusions (IUTs) of red blood cells are a life-saving intervention for fetal hemolytic disease resulting from Rhesus alloimmunization and have become the standard treatment for various causes of fetal anemia [26]. Although randomized trials have not been conducted, observational studies indicate that IUTs significantly improve survival rates in cases of severe fetal anemia due to parvovirus B19 infection. Specifically, IUT has reduced mortality rates from approximately 50% to 18%, with most non-hydropic fetuses—those without an abnormal accumulation of fluid in two or more fetal compartments—typically requiring only a single transfusion [9,10]. A meta-analysis involving 14 studies and 1436 cases of fetal parvovirus B19 infection reported a survival rate of 82% for transfused fetuses compared to 55% for those who did not receive transfusions [10]. The effectiveness of middle cerebral artery peak systolic velocity (MCA PSV) in predicting anemia during pregnancies undergoing IUT appears to be declining; however, ongoing monitoring with MCA PSV is essential until the risks of post-procedural complications and recurrent anemia are adequately addressed [27]. The timing of IUT depends on the expertise and capabilities of the treatment center, as well as individual clinical factors, such as placental location and the techniques employed. IUT can be performed before 20 weeks’ gestation via intravascular or intraperitoneal routes and has even been conducted as early as the first trimester [10,28]. The initiation of intensive fetal infection monitoring will be tailored according to the gestational age at which the center can offer IUT. Thrombocytopenia may also occur alongside severe anemia in this viral infection, potentially increasing the risk of bleeding associated with IUT [10,24]. Although a concurrent platelet mass transfusion can be administered, potential complications include fluid overload, thrombosis, and heart failure. These risks must be balanced against the overall low incidence of fetal hemorrhagic complications associated with parvovirus B19 infection, while also considering the severity of any thrombocytopenia [8,10,24].

Postpartum, no specific investigation is required in newborns in whom no signs of fetal anemia or other sonographic features of congenital parvovirus infection have been identified. Newborns with fetal anemia requiring IUT, myocarditis, or hydrops should undergo regular pediatric examination and follow-up [9].

## 9. Should Parvovirus B19 Infection Be Included in Routine Screening of Pregnant Patients?

The question of routine parvovirus B19 screening for pregnant women is complex. While the virus can be dangerous for the fetus, it is usually mild, and widespread screening has downsides. Currently, no international guidelines recommend screening all pregnant women. This is because of the following reasons:Low Risk: Most infections are mild or asymptomatic, and serious complications are rare.No Prevention: There is no vaccine or treatment to prevent infection or transmission to the fetus.Costly: Universal screening would be expensive and may not be a good use of resources.Logistical Challenges: Screening and monitoring require significant healthcare infrastructure.False Positives: Inaccurate test results can cause unnecessary anxiety and invasive procedures.

However, targeted screening may be beneficial for the following:High-Risk Occupations: Healthcare workers, daycare providers, and teachers have increased exposure risk.Previous Complications: Women with a history of parvovirus-related pregnancy problems.Outbreaks: Screening may be useful in areas with high prevalence during outbreaks [9,10,17].

Recommendations for Romania:

Given Romania’s climate and the seasonal nature of parvovirus B19, targeted screening is recommended for high-risk occupational groups and women with a history of complications. Public health campaigns should raise awareness and promote preventive measures. Further research is needed to determine the prevalence of the virus in Romania and the cost-effectiveness of different screening strategies.

Limitations of Current Guidelines:

Current guidelines often rely on limited evidence and do not always consider regional variations in prevalence. Emerging diagnostic tools, like rapid point-of-care tests, could improve screening feasibility in the future [14,17,23].

Balancing Benefits and Risks:

While early detection is important, it is crucial to minimize anxiety caused by false positives. Shared decision-making between healthcare providers and pregnant women is essential [10,11,20].

Alternative Approaches:

Selective screening based on risk factors or outbreaks may be more efficient. For example, France has successfully implemented selective screening for pregnant women working in childcare [10,20,22,23].

Future Directions:

More research is needed to determine the usefulness of PCR screening in asymptomatic pregnancies and to assess the long-term effects of maternal infection on children’s development [10,16,17].

The Canadian Task Force on Preventive Health Care and the American College of Obstetricians and Gynecologists provide the following recommendations for screening for parvovirus B19 infection [9,16,21,22]:Investigation for parvovirus B19 infection is advised as part of standard screening for hydrops fetalis or intrauterine fetal death (II-2A).Routine testing for parvovirus immunity in low-risk pregnancies is not recommended (II-2E).Pregnant women who have been exposed to or exhibit symptoms of parvovirus B19 infection should be assessed to determine their susceptibility to infection (non-immune) or whether they have an active infection by evaluating their immunoglobulin G and immunoglobulin M levels for parvovirus B19 (II-2A).If immunoglobulin G for parvovirus B19 is detected and immunoglobulin M is negative, the woman is considered immune and can be reassured that she will not develop the infection and that the virus will not adversely affect her pregnancy (II-2A).If both immunoglobulin G and immunoglobulin M for parvovirus B19 are negative (and the incubation period has passed), the woman is not immune and has not contracted the infection. She should be advised to minimize exposure at work and at home, with decisions regarding absence from work made on a case-by-case basis (II-2C). Further research is encouraged to explore strategies for minimizing exposure, particularly concerning occupational risks (III-A).If a woman is diagnosed with a recent parvovirus B19 infection, a referral to an obstetrician or maternal–fetal medicine specialist should be considered (III-B). She should be counseled regarding the risks of fetal transmission, fetal loss, and hydrops. Serial ultrasounds should be conducted every 1–2 weeks for up to 12 weeks post-infection to monitor for signs of anemia (using Doppler measurement of maximum middle cerebral artery systolic velocity) and hydrops (III-B). If hydrops or evidence of fetal anemia is detected, a referral should be made to a specialist capable of performing fetal blood sampling and intrauterine transfusions (II-2B).

## 10. Conclusions

In conclusion, routine screening for parvovirus B19 infection can be considered in communities where infection is common, there is occupational exposure, or during endemic periods, as accurate identification and treatment of fetuses affected by congenital B19V infection have been shown to improve perinatal outcomes [9,10]. After exposure, the clinical management of parvovirus B19 infection during pregnancy focuses on early fetal monitoring for anemia and hydrops development. Serial ultrasound examinations are recommended to assess fetal well-being, and referral to a maternal–fetal specialist is crucial if hydrops is identified. For fetuses with severe anemia, intrauterine transfusion (IUT) remains the gold standard for intervention and improving fetal outcomes [5,10]. While significant advancements have been made in understanding and managing parvovirus B19 infection during pregnancy, further research is needed to address unanswered questions regarding optimal diagnostic methods and management strategies, and to determine whether routine screening should be offered to all pregnant patients.

## Figures and Tables

**Figure 1 life-14-01667-f001:**
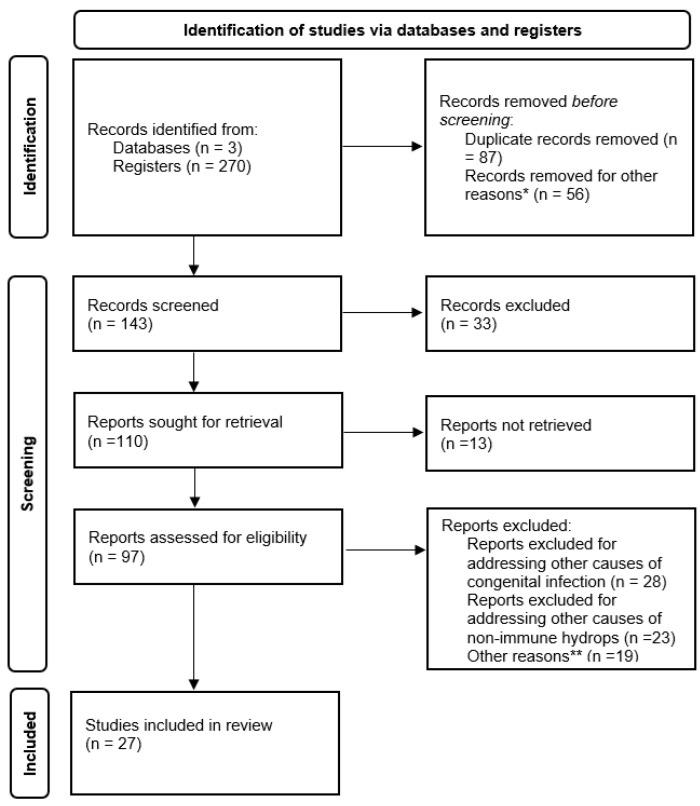
PRISMA 2020 diagram for article inclusion. * irrelevant topic, wrong study design, or inability to access the full text. ** not meeting the inclusion criteria, poor quality, or duplicates that were missed in the initial screening.

**Figure 2 life-14-01667-f002:**
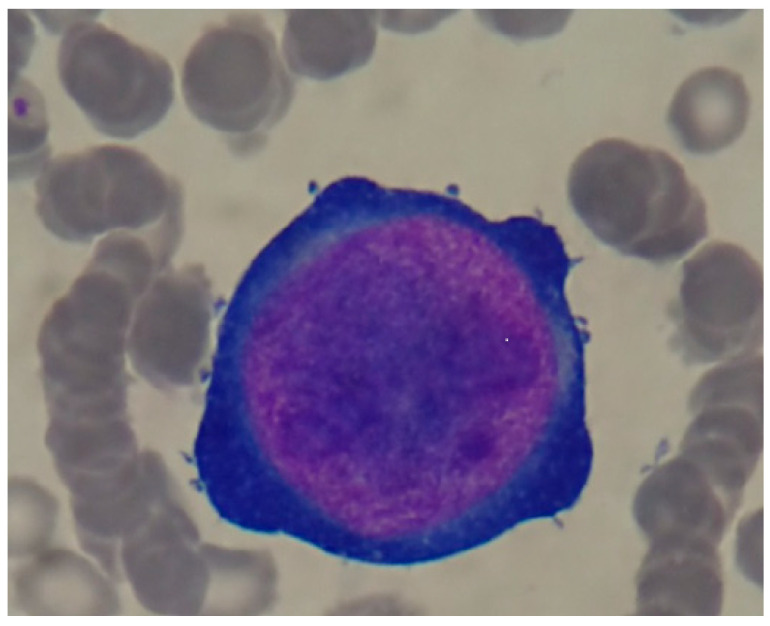
Giant proerythroblasts [12] (https://www.researchgate.net/figure/Giant-proerythroblasts-with-intranuclear-eosinophilic-inclsionbodies-like-nucloli_fig1_376949624, Accessed on 5 December 2024).

**Figure 3 life-14-01667-f003:**
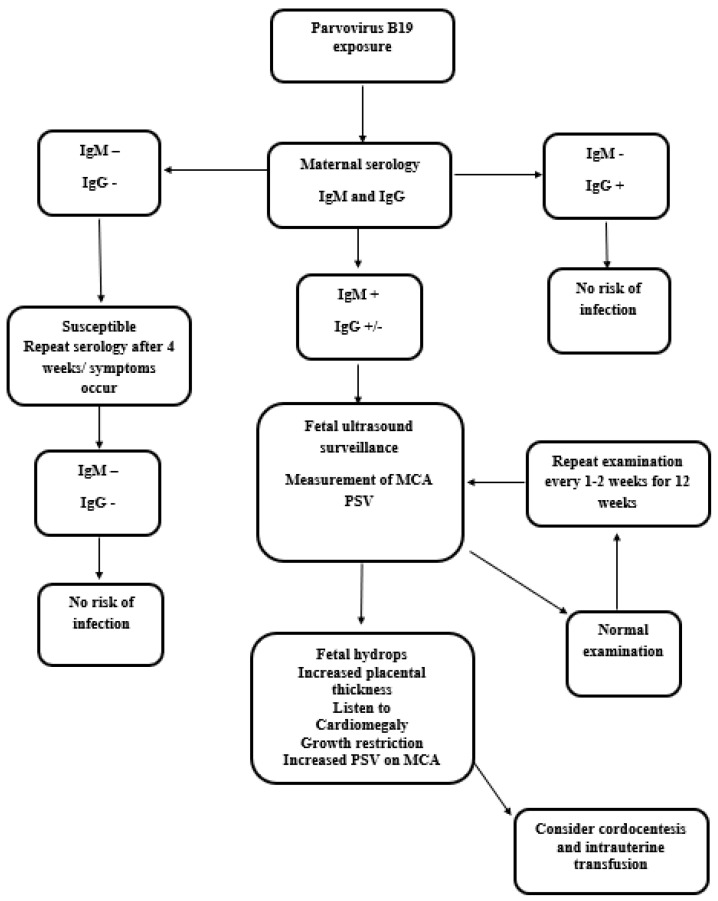
Diagnosis and management of patients exposed to parvovirus B19 infection [9,10].

**Figure 4 life-14-01667-f004:**
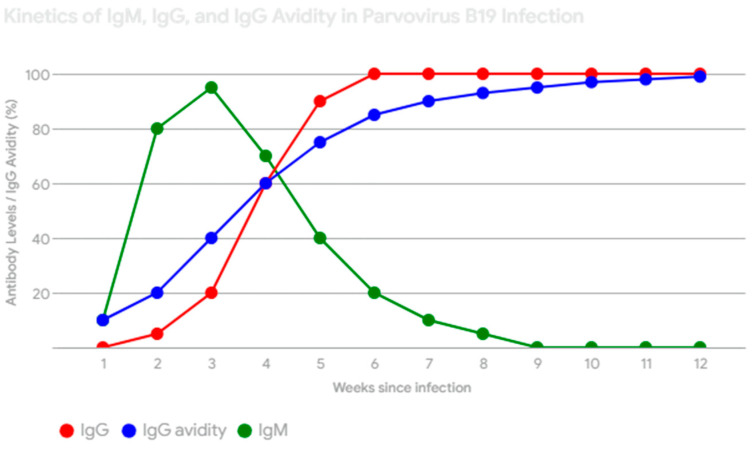
Kinetics of IgM and IgG against B19v infection an IgG avidity [10,18,19,20]. This figure shows the typical antibody response, including IgM and IgG levels and IgG avidity changes over time. IgM (green) appears first, followed by IgG (red). IgG avidity (blue) increases over time, reflecting immune response maturation. Low avidity suggests a recent infection, while high avidity indicates a past infection [9,10,21,22].

**Table 1 life-14-01667-t001:** Possible maternal–fetal effects of parvovirus B19 infection.

Maternal Asymptomatic Erythema infectiosum Arthropathy Anemia Myocarditis	Fetal Fetal death Anemia/Hydrops Myocarditis

## Data Availability

The original contributions presented in the study are included in the article; further inquiries can be directed to the corresponding author/s.

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
