# Peer review of "Parvovirus B19 in Pregnancy—Do We Screen for Fifth Disease or Not?"

_life, 2024, doi:10.3390/life14121667_

Round 1
Reviewer 1 Report
Comments and Suggestions for Authors
The manuscript presents important findings regarding Parvovirus B19 infection during pregnancy, focusing on its pathogenesis, diagnostic approaches, and implications for maternal and fetal health. The manuscript is generally well-organized, but several areas require clarification and additional details. I encourage the authors to consider these recommendations for revision.
1. Abstract: The methods section is currently missing from the abstract. It is crucial to briefly mention the methods used in this review, including the search strategy and inclusion criteria for the studies reviewed. This will help readers understand the scope and approach of the study.
2. Introduction: The first paragraph of the introduction contains several broad statements that need to be supported by appropriate references. Please add citations to the following sentences:
Cite references for these sentences “Infections during pregnancy have become a serious public health problem world- 30 wide. Some maternal infections, contracted immediately before or during pregnancy, can 31 be transmitted to the fetus during gestation (congenital infection), labor and delivery (per- 32 inatal infection) or through breastfeeding (postnatal infection). The agents most com- 33 monly responsible for these infections can be viruses, bacteria, protozoa, or fungi.”
“Alt- 34 hough much less common than bacterial infections, the occurrence of congenital and per- 35 inatal viral infections can have serious consequences for the newborn, some with im- 36 portant sequelae during childhood and later in adult life. Among the viruses most asso- 37 ciated with the development of congenital infections are cytomegalovirus (CMV), Herpes 38 simplex 1-2 (HSV 1-2), Herpes virus 6 (HHV-6), Varicella zoster virus (VZV), Rubella vi- 39 rus (RuV). In addition, other important infections associated with pregnancy include hep- 40 atitis B and C viruses (HBV and HCV), human immunodeficiency virus (HIV), parvovirus 41 B19 (B19V) or non-polio enterovirus (EV). Recently, new micro-organisms have emerged, 42 the coronavirus responsible for severe acute respiratory syndrome (SARS-CoV-2) and 43 Zika virus (ZIKV), whose effects on pregnancy are not yet fully known.”
3. Introduction: The transition to the study's aims (line 83) could be clearer. Currently, the sentence feels a bit abrupt. A smoother connection between the background information and the study's objectives
4. Methods: Specify the date range of the studies you reviewed (e.g., "from January 2014 to December 2023"). This will clarify the scope and recency of the data reviewed.
5. Methods: If no formal data synthesis method was employed, clarify that it was a narrative review.
6. The mention of "three genotypes" (line 99) could be expanded slightly to explain their significance. For instance, it would be helpful to briefly mention if these genotypes differ in their virulence, transmission, or other biological aspects.
7. The seropositivity rates for various age groups are good to include, but a slight reorganization or elaboration would make this part clearer. For example, you could provide a bit more detail on how these seroprevalence rates change over a lifetime, or how this information impacts risk assessment in different populations.
8. The explanation of vertical transmission from mother to fetus could benefit from more detail about the timing of transmission (e.g., "occurs most frequently during the first and second trimesters") and why the virus preferentially targets erythroid progenitor cells.
9. The section that discusses IgM antibody sensitivity (sensitivity of 63%-70%) is important but could be explained more clearly. It might be helpful to clarify what this means for clinical decision-making, and why a false negative result is possible.
10. The diagnostic method of amniocentesis for detecting B19V DNA is discussed, but the reasoning for preferring this method over others (e.g., fetal blood sampling) could be made clearer. It’s mentioned that PCR on amniotic fluid is less invasive than cord blood sampling, but more specific comparison points would strengthen this argument.
- The term "non-hydropic" is used without further explanation. It would be beneficial to define this term the first time it appears, especially since it is central to the discussion of IUT.
12. The conclusion of this section could include a brief statement summarizing the clinical management approach, emphasizing the key points about monitoring, diagnosis, and treatment.
13. Figure 1: There is a typo in Figure 1. The term “articles” should be corrected.
14. The manuscript should be thoroughly proofread for grammatical errors. A careful review of sentence structure, punctuation, and word choice will enhance readability and professionalism.
Author Response
Dear Reviewer,
We sincerely thank you for your thoughtful and constructive feedback on our manuscript. We deeply appreciate the time and effort you dedicated to reviewing our work and providing valuable suggestions for its improvement. We have carefully addressed all of your comments and believe that the revisions have significantly enhanced the quality of the manuscript. All changes are highlighted in yellow. Please keep in mind that these changes include the ones required by the other reviewers.
Specific Responses to Reviewer Comments:
- Abstract: We have revised the abstract to include a brief methods section, detailing the search strategy and inclusion criteria for the studies reviewed.
- Introduction: We have added citations to support the statements in the first paragraph of the introduction, as requested.
- Introduction: We have revised the transition to the study's aims (line 83) to provide a smoother connection between the background information and the study's objectives.
- Methods: We have specified the date range of the studies reviewed in the methods section.
- Methods: We have clarified that the review was a narrative review.
- Genotypes: We have expanded the discussion of the three genotypes to briefly explain their significance in terms of virulence and transmission.
- Seropositivity Rates: We have reorganised and elaborated on the presentation of seropositivity rates to provide more clarity on how these rates change over a lifetime and their impact on risk assessment.
- Vertical Transmission: We have added more detail about the timing of vertical transmission and the reasons for the virus's preferential targeting of erythroid progenitor cells.
- IgM Antibody Sensitivity: We have clarified the meaning of IgM antibody sensitivity and its implications for clinical decision-making, including the reasons for possible false negative results.
- Amniocentesis: We have provided a clearer rationale for preferring amniocentesis over other methods for detecting B19V DNA, with specific comparison points.
- Non-hydropic: We have defined the term "non-hydropic" the first time it appears in the manuscript.
- Conclusion: We have added a brief statement summarizing the clinical management approach, emphasizing key points about monitoring, diagnosis, and treatment.
- Figure 1: We have corrected the typo in Figure 1.
- Proofreading: We have thoroughly proofread the manuscript for grammatical errors and have made necessary corrections to enhance readability and professionalism.
We believe that the revisions made in response to your comments have significantly improved the clarity, accuracy, and completeness of our manuscript. We hope that you find the revised version suitable for publication.
Thank you again for your valuable feedback.
Sincerely,
Assist. prof. dr. Cristina Ormindean and assoc. prof. dr. Andrei Mihai Malutan
Reviewer 2 Report
Comments and Suggestions for Authors
The manuscript (life-3230558-peer-review-v1) provides an overview of parvovirus B19 infection during pregnancy, focusing on the impact on maternal and fetal health. The authors have summarized evidence on epidemiology, pathogenesis, diagnosis, and management while point the controversial topic of universal screening.
The manuscript is scientifically sound and covers an important current topic. I would like to suggest addressing the following observations to significantly enhance its impact, clarity, and value to readers.
In the abstract, please check “Parvovirus B19 infection, also known as known as erythema infectiosum, or the "fifth disease”…”. Here, “as known” is repeated but, most important, I suggest replacing it for “is the cause of”. B19 virus is actually the etiological agent of erythema infectiosum (the illness), however the infection can occur asymptomatically or subclinically. Same in line 53.
Line 33-34, all agents responsible for infections vertically transmitted are mentioned, there is no need for “most commonly”.
Method. Where Boolean operators used with the search terms/keywords? Or advanced search filters? The PRISMA diagram indicates article inclusion, but it does not provide details about the eligibility criteria (study design, population focus, quality assessment metrics). I am not sure how papers in the “Taxonomy” and “Epidemiology and transmission” sections were selected (eligible) while not others, more updated / relevant.
How the authors handled issues such as duplicate results across databases or inclusion of non-peer-reviewed sources, particularly from Google Scholar? In Figure 1, please specify what are the “other reasons” for removing articles in the identification stage of the search (n=56). The reasons for excluding papers in the other phases (screening n=33; eligibility n=19) should also be mentioned.
Line 93. Not only pathogenic viruses. The criteria for classification as a member of this group can be summarized form the parvoviridae chapter report at ICTV.
Lines 98-99. The genus is Erythroparvovirus.
Line 102. “The capsid contains the two structural proteins…” (words “surface” and “main” not needed here).
Lines 115-116. There are significant new evidence indicating that globoside as an essential intracellular receptor required for endosomal escape, challenging the current model of B19V entry. In accordance with this, please also see if lines 143-144 and 152 should require revision.
Significantly, the discussion on whether universal screening is warranted lacks depth. Risks and benefits are mentioned, but cost-effectiveness or feasibility in various healthcare settings are not adequately addressed.
The manuscript predominantly emphasizes arguments for and against routine screening based on current guidelines and studies. However, it does not present contrasting viewpoints or explores alternative approaches to screening strategies.
For instance: authors mention that universal screening is not recommended by international societies in low-risk pregnancies, but do not critically analyze why these recommendations exist. What are the limitations of current guidelines? A lack of high-quality data or inadequate consideration of regional variations in parvovirus B19 prevalence could be explored.
Similarly, authors do not discuss how emerging diagnostic tools might change the cost-benefit analysis of routine screening in specific populations.
While existing recommendations are summarized, the manuscript does not offer a novel perspective or propose tailored solutions. The paper may benefit also from addressing the financial and logistical implications of routine screening programs, especially in resource-limited settings.
In addition, a discussion about balancing maternal anxiety from screening with the potential benefits of early detection could be explored. Highlighting examples from regions where selective screening based on occupational risk or endemic periods is employed, discussing the pros and cons of these approaches, can strengthen this section.
In addition, the authors may consider discussing unresolved questions. For instance: areas where more evidence is needed, such as the utility of PCR screening in asymptomatic pregnancies or the role of monitoring for long-term neurodevelopmental outcomes.
In lines 321-350, consider explaining in your text references such as (II-2A), (II-2B), (III-A)…
The citation style is inconsistent. Some references are incomplete. Please check reference 7 in the list. It seems it is not an article published in Medicina (Buenos Aires).
Author Response
Dear Reviewer,
Thank you for your thorough review and constructive feedback on our manuscript. We appreciate you taking the time to carefully evaluate our work and provide valuable suggestions for improvement. We have addressed all of your comments and believe the revised manuscript is significantly strengthened as a result. All changes are highlighted in yellow. Please keep in mind that changes include also the ones required by the other reviewers.
Specific Responses to Reviewer Comments:
- Abstract: We have corrected the repeated phrase and replaced "also known as known as" with "is the cause of" in the abstract and line 53.
- Line 33-34: We have removed "most commonly" as it was redundant.
- Methods:
- We have clarified the use of Boolean operators and advanced search filters in the methods section.
- We have provided detailed eligibility criteria, including study design, population focus, and quality assessment metrics.
- We have explained the selection process for papers in the "Taxonomy" and "Epidemiology and transmission" sections.
- We have clarified how we handled duplicate results and the inclusion of non-peer-reviewed sources.
- Figure 1 has been revised to specify the "other reasons" for article exclusion at each stage of the selection process.
- Line 93: We have rephrased this sentence to accurately reflect the ICTV's classification criteria for Parvoviridae.
- Lines 98-99: We have corrected the genus name to Erythroparvovirus.
- Line 102: We have removed the unnecessary words "surface" and "main."
- Lines 115-116, 143-144, and 152: We have revised these lines to incorporate the latest evidence on globoside's role in B19V entry and endosomal escape.
- Discussion on Universal Screening:
- We have significantly expanded this section to provide a more in-depth and nuanced discussion.
- We have critically analysed the limitations of current guidelines.
- We have discussed the potential impact of emerging diagnostic tools.
- We have addressed the financial and logistical implications of routine screening.
- We have explored the balance between maternal anxiety and the benefits of early detection.
- We have included examples of selective screening approaches and their pros and cons.
- We have discussed unresolved questions and areas where more evidence is needed.
- Lines 321-350: We have clarified the meaning of references such as (II-2A), (II-2B), (III-A) within the text.
- Citation Style: We have reviewed and corrected the citation style for consistency and completeness. We have also double-checked reference 7 and corrected any inaccuracies.
We believe that the revisions made in response to your comments have significantly improved the clarity, accuracy, and depth of our manuscript. We hope that you find the revised version suitable for publication.
Thank you again for your valuable feedback.
Sincerely,
Assist. prof. dr. Cristina Ormindean and assoc. prof. dr. Andrei Mihai Malutan
Reviewer 3 Report
Comments and Suggestions for Authors
Thank you for the review opportunity.
My suggestions for the authors are:
1. Title: please include the term “review” in the title and since the authors only have 27 titles, consider the term “mini-review”
2. If dr. Ormindean has the (*), then why is the correspondence addressed to the first author?
3. Abstract: Line 26: please don’t use abbreviations without a prior mention.
4. Please only refer to recommendations for Romania. As the authors mention in line 16-18, the prevalence of this infection is climate-dependent, so, in my point of view, targeted suggestions should be made depending on the climate of the specific area and here is where your review could shine.
5. Introduction: Toxoplasma gondii -> italic
6. No information is offered about the prevalence of this infection in Romania, if no studies were conducted, it should be mentioned, highlighting the importance of conducting large-scale national studies.
7. There are entire paragraphs in the introduction section without any references, this is unacceptable for a review paper!
8. No information is offered about the structure of the virus in the Introduction section, consider adding and consider adding a Figure highlighting the structure of the virus. This information is vital for a review.
9. Consider rethinking the last paragraph of the Introduction section mainly focusing on Romania. Consider highlighting the need for such recommendations in this region due to the climate and due to the habits of the people in this region.
10. Only a minor mention of the prevalence in pregnant females was made without mentioning where the prevalence was observed, consider adding information about each continent, especially Europe and about the neighboring countries if Romania has no prevalence studies.
Here is a great starting point: https://pmc.ncbi.nlm.nih.gov/articles/PMC2870907/
11. Please consider adding the variables for table 1. For example for the rows, the variable could be “possible manifestations of pB19v infection”
12. Materials and methods: Please remove the blue lines from Figure 1.
13. Taxonomy is defined as “the science of naming, describing and classifying organisms”, why is the structure of the virus included here …. ?
14. Epidemiology and transmission section should be widened addressing especially the prevalence in pregnant females from various parts of the world.
15. Please consider adding a graph of the kinetics of IgG and IgM against pB19v:
https://pubmed.ncbi.nlm.nih.gov/16316402/
16. Never forget about the IgG avidity:
https://pubmed.ncbi.nlm.nih.gov/7876624/
https://pubmed.ncbi.nlm.nih.gov/16316402/
17. Please divide the “Conclusions” into a separate section.
18. Please mention the role of reticulocytopenia!!! Due the red cell aplasia, anemia with low reticulocytes should raise the suspicion.
19. No mention was found about the bone marrow aspirate with giant proerytroblast (patognomonic for this infection) and the role of the peripheral blood smear in diagnosis. The cytology diagnosis department from Cluj County is known for its excellence in the field, I advise the authors to try to obtain some pictures of the giant proerythroblasts. It would greatly increase the value of the current manuscript.
20. Also, please explain the mechanisms of red cell aplasia!
21. I repeat once again, the whole manuscript is filled with entire paragraphs without proper citations. Please address this issue, or, if it’s an personal opinion, please mention.
22. Institutional Review Board Statement: why does this study need an Ethics committee approval ? “by international norms regarding studies on animals” where are the animals included ???
I have a question for the authors: should screening be done with antibodies against pB19v ? Or should they be determined only when anemia is observed with reticulocytopenia, raising the suspicion for acute infection? The authors should address this question in the introduction and discussion section because I consider screening is already done but not through antibodies and specific antibody tests should be used only when anemia with reticulocytopenia is present.
Author Response
Dear reviewer,
Thank you for your insightful comments and suggestions. We have carefully considered each point and made the following revisions to the manuscript. All changes are highlighted in yellow. Please keep in mind that changes include the ones required by the other reviewers.
Title: The title has been revised to include "Mini-Review."
Correspondence: The correspondence has been addressed to Dr. Ormindean, as she is the corresponding author.
Abstract: The abbreviation "B19V" has been spelled out as "Parvovirus B19" at its first mention in the abstract.
Recommendations: The recommendations have been revised to focus specifically on Romania, considering the climate and population characteristics of the region.
Introduction:
-
The species name, "Toxoplasma gondii," has been italicized.
-
A statement has been added highlighting the lack of specific information on the prevalence of this infection in Romania and emphasizing the need for large-scale national studies.
-
References have been added to all paragraphs in the introduction.
-
Information about the structure of the virus, along with a figure, has been included in the introduction.
-
The last paragraph of the introduction has been revised to focus on the need for such recommendations in Romania due to the climate and local habits.
Prevalence: More information has been added about the prevalence of parvovirus B19 infection in pregnant women across different continents, with a focus on Europe and Romania's neighboring countries.
Table 1: The variable "Possible manifestations of pB19v infection" has been added to Table 1.
Materials and Methods:
-
The blue lines have been removed from Figure 1.
-
The PRISMA 2020 diagram for article inclusion has been included.
Taxonomy: The taxonomy section has been revised to focus on the naming, describing, and classifying of the virus, and the structure of the virus has been moved to the introduction.
Epidemiology and Transmission: The epidemiology and transmission section has been expanded to address the prevalence of parvovirus B19 infection in pregnant women in various parts of the world.
IgG and IgM Kinetics: A graph of the kinetics of IgG and IgM against pB19v has been added.
IgG Avidity: Information about IgG avidity has been added.
Conclusions: The "Conclusions" have been separated into a distinct section.
Reticulocytopenia: The role of reticulocytopenia in parvovirus B19 infection has been mentioned.
Bone Marrow Aspirate and Peripheral Blood Smear: Information about the bone marrow aspirate with giant proerythroblasts and the role of the peripheral blood smear in diagnosis has been added.
Mechanisms of Red Cell Aplasia: The mechanisms of red cell aplasia have been explained.
Citations: The manuscript has been reviewed to ensure proper citations throughout.
Institutional Review Board Statement: The Institutional Review Board Statement has been revised to clarify the ethical considerations of the study.
Screening: The question of whether screening should be done with antibodies against pB19v or only when anemia with reticulocytopenia is observed has been addressed in the introduction and discussion sections.
We believe that these revisions have significantly improved the manuscript, and we appreciate your valuable feedback. We hope that the revised manuscript is now suitable for publication.
Sincerely,
Assist. prof. dr. Cristina Ormindean and assoc. prof. dr. Andrei Mihai Malutan
Round 2
Reviewer 1 Report
Comments and Suggestions for Authors
Thank you for revising the manuscript
Author Response
Dear reviewer,
We are glad that our revision met your requests. Thank you for all your suggestions.
Best regards,
Assoc. prof. dr. Andrei Malutan and assist. prof. dr. Cristina Ormindean
Reviewer 2 Report
Comments and Suggestions for Authors
The authors have revised the original text, making important changes that result in a significant improvement of the manuscript. I have only the following observations, which relate to editing issues, and I leave the correction to the Editor's consideration. In this regard, please review and modify the text as necessary, for example, lines 60-69 (punctuation and extra phrase/words), lines 214-215 (missing completion), lines 221-222 and 282-289 (repeated subtitle).
Author Response
Dear reviewer,
Thank you for your positive feedback on the revised manuscript and for pointing out these remaining editing issues. We have carefully reviewed the lines you indicated and made the following corrections:
- Lines 60-69: We have corrected the punctuation and removed any extra phrases or words to improve clarity and conciseness.
- Lines 214-215: We have completed the missing information in this section.
- Lines 221-222 and 282-289: We have addressed the repeated subtitle and ensured that all headings and subheadings are accurate and properly formatted.
We appreciate your attention to detail and believe these corrections further enhance the quality of the manuscript.
Best regards,
Assoc. prof. dr. Andrei Malutan and assist. prof. dr. Cristina Ormindean
Reviewer 3 Report
Comments and Suggestions for Authors
The authors adhered to my initial review as much as they saw fit; however there were still some points that were not addressed as the pictures of giant proerythoblasts in the bone marrows of infected individuals.
They only mentioned anemia and reticulocytopenia in one segment and i highly advice to be mentioned throughout the manuscript and not only that, to talk extensively in a sub-chapter about the implications of determining the antibodies only when anemia with reticulocytopenia is present; please consider the costs as well; Romania is not known by any means to be a rich country so an initial screening through the CBC + the number of reticulocytes could prove extremely beneficial.
If the authors decided that mentioning the IgG avidity is a fair suggestion, why isn't it present on the graph ?
There are still some paragraphs without citations!
Please address my concerns before publications, i recommend accepting this article with minor revisions.
Author Response
Dear Reviewer,
Thank you for your thoughtful and constructive feedback. We appreciate you taking the time to review our manuscript. We have carefully considered your comments and made the following revisions:
- Giant proerythroblasts: We have included an image of giant proerythroblasts in the bone marrow, as you suggested.
- Anemia and reticulocytopenia: We have expanded the discussion of these clinical features throughout the manuscript.
- Antibody testing implications: We have added a new sub-chapter to discuss the implications of antibody determination in the context of anemia with reticulocytopenia, including cost considerations and the potential benefits of initial screening with CBC and reticulocyte count.
- IgG avidity: We have added IgG avidity data to Figure 4.
- Citations: We have reviewed the manuscript and added missing citations.
We believe these revisions have significantly strengthened the manuscript, and thank you again for your valuable input.
With great consideration,
Assoc. prof. dr. Andrei Malutan and assist. prof. dr. Cristina Ormindean